# Leptin and Associated Mediators of Immunometabolic Signaling: Novel Molecular Outcome Measures for Neurostimulation to Treat Chronic Pain

**DOI:** 10.3390/ijms20194737

**Published:** 2019-09-24

**Authors:** Thomas M. Kinfe, Michael Buchfelder, Shafqat R. Chaudhry, Krishnan V. Chakravarthy, Timothy R. Deer, Marc Russo, Peter Georgius, Rene Hurlemann, Muhammad Rasheed, Sajjad Muhammad, Thomas L. Yearwood

**Affiliations:** 1Department of Neurosurgery, Division of Functional Neurosurgery and Stereotaxy, University of Erlangen-Nueremberg, 91054 Erlangen, Germany; Michael.Buchfelder@uk-erlangen.de; 2Departmen of Basic Medical Sciences Shifa College of Pharmaceutical Sciences, Shifa Tameer-e-Millat University, Islamabad 44000, Pakistan; shafqatrasul@yahoo.com; 3Department of Anesthesiology and Pain Medicine, University of California, San Diego, CA 92161, USA; kvchakravarthy@UCSD.EDU; 4VA San Diego Healthcare System, California, San Diego, CA 92161, USA; 5The Spine and Nerve Center of the Virginias, Charleston, WV 25301, USA; doctdeer@aol.com; 6Hunter Pain Clinic, 2300 Newcastle, Australia; algoguy@gmail.com; 7Nambour Selangor Private Hospital, Nambour, 4560 Queensland, Australia; painrehab@netspace.net.au; 8Department of Psychiatry, Division of Medical Psychology, University Hospital Bonn, 53127 Bonn, Germany; rene.hurlemann@ukb.uni-bonn.de; 9Department of Neuro-Urology Marien Hospital Herne, University Clinic of the Ruhr University Bochum, 44625 Herne, Germany; muhammad.rasheed@ruhr-uni-bochum.de; 10Department of Neurosurgery, University Hospital Duesseldorf, 40225 Düsseldorf, Germany; msajjad51@yahoo.com; 11Neuromodulation Specialists LLC, Fairhope, AL 36532, USA; nopaindr@mac.com

**Keywords:** chronic pain, immunometabolism, systemic inflammation, neuromodulation, biomarkers development, quantitative outcome measures

## Abstract

Chronic pain is a devastating condition affecting the physical, psychological, and socioeconomic status of the patient. Inflammation and immunometabolism play roles in the pathophysiology of chronic pain disorders. Electrical neuromodulation approaches have shown a meaningful success in otherwise drug-resistant chronic pain conditions, including failed back surgery, neuropathic pain, and migraine. A literature review (PubMed, MEDLINE/OVID, SCOPUS, and manual searches of the bibliographies of known primary and review articles) was performed using the following search terms: chronic pain disorders, systemic inflammation, immunometabolism, prediction, biomarkers, metabolic disorders, and neuromodulation for chronic pain. Experimental studies indicate a relationship between the development and maintenance of chronic pain conditions and a deteriorated immunometabolic state mediated by circulating cytokines, chemokines, and cellular components. A few uncontrolled in-human studies found increased levels of pro-inflammatory cytokines known to drive metabolic disorders in chronic pain patients undergoing neurostimulation therapies. In this narrative review, we summarize the current knowledge and possible relationships of available neurostimulation therapies for chronic pain with mediators of central and peripheral neuroinflammation and immunometabolism on a molecular level. However, to address the needs for predictive factors and biomarkers, large-scale databank driven clinical trials are needed to determine the clinical value of molecular profiling.

## 1. Introduction

The estimated prevalence of chronic pain varies between 20–40% and is characterized by an increased socioeconomic burden negatively affecting both the physical condition and quality of life of patients [1]. Chronic pain conditions pose very high costs of treatment worldwide. The estimated costs (National academic press, USA) of pain treatments ranged between US$ 560–635 billion in 2010. Clearly, the recent opioid crisis demonstrated the need to replace former treatment strategies for chronic pain disorders with non-opioid pharmacological paradigms and neurostimulation therapy.

Chronic pain, as a consequence of tissue injury, is characterized by activation and sensitization of central and peripheral pain pathways mediated through multiple inflammatory molecules including leptin, its relatives adiponectin and ghrelin, interleukin (IL) IL1-β, tumor necrosis factor TNF-α, IL-6, high-mobility group box-1 HMGB-1, damage-associated molecular pattern DAMP´s, oxytocin, and multiple chemokines, reactive oxygen species, ATP, lipids, and vasoactive amines. Neuropathic pain is initiated by a lesion in the somatosensory nervous system, is strongly associated with neuroinflammation, and is considered a more severe form of pain that also influences sleep quality and psychological functioning [2]. Experimental studies suggest that these inflammatory peptides including adipocytokines interact with neural transmission by promoting hyperexcitability in neural circuits relevant for the development of chronic pain disorders (e.g., spinal cord dorsal column, dorsal root ganglia, and trigeminal ganglion). For instance, Il-1 β and leptin (pro-inflammatory) and oxytocin (anti-inflammatory) exert their effects through specific receptors located either at the terminal C-fibers (neuropathic pain) or the trigeminal ganglion (migraine). Pharmacological treatment with a variety of classes of drugs and behavioral therapy are first-line treatments for neuropathic pain [3,4,5,6,7,8], but many patients fail to experience clinically meaningful pain relief from these treatments alone. Drug resistance and loss of efficacy in chronic pain treatment are perhaps due to the involvement of multiple disturbed physiological and psychological interrelated pathways that contribute to the severity of chronic pain. Hence, understanding mechanisms of relevant comorbidities underlying chronic pain (e.g., metabolism, depression, and anxiety) may help to develop novel diagnostic and treatment options. In addition to their role in the transition and maintenance of chronic pain, most of the described mediators have been associated with a broad range of chronic disorders (neurological, psychiatric, cardiovascular, and metabolic) due to their multifunctional characteristics, which in turn may explain the clinically observed co-occurrence of the so-called comorbidities [9,10,11,12]. Spinal and peripheral neurostimulation therapies have repeatedly shown meaningful pain relief in recent studies to treat various chronic pain disorders (failed back surgery, complex regional pain syndrome (CRPS), and migraine) targeting the spinal cord, the dorsal root ganglia (DRG), and peripheral nerves (vagus nerve stimulation; VNS) [3,4,5,6,7,8]. The strongest rationale for considering neurostimulation approaches to treat chronic pain is the growing understanding that all forms of pain are ultimately the result of an interplay of central and peripheral properties of the nervous system, which in turn allows neurostimulation therapies to interact with inflammatory pain signaling [13,14,15,16,17,18]. Although psychometric alterations (depression, cognitive decline, and anxiety) and functional parameters such as sleep or metabolic state have been reported to interact with neurostimulation responsiveness, objective and quantitative outcome measures on a molecular level remain largely under-investigated. The majority of published studies assess neurostimulation outcomes using subjective score-based tools [18]. In this narrative review, we provide experimental evidence and describe the current state of in-human studies assessing possible relationships between the responsiveness of neurostimulation therapies for chronic pain and molecular profiling of circulating mediators relevant for functional metabolic homeostasis including leptin and its relatives adiponectin and ghrelin, and other associated mediators of metabolic-related inflammation.

## 2. Methods

Search strategies encompassed published literature identified through searches of PubMed, MEDLINE/OVID, and SCOPUS and manual searches of the bibliographies of original and review articles from 2000–2019 with the following terms: chronic pain disorders, systemic inflammation, immunometabolism, adipokines, cytokines, chemokines, biomarkers, metabolic disorders, neuromodulation for chronic pain, and in-human studies to identify primary outcome measures using molecular profiling of inflammation. The primary criteria for inclusion in our review were the following terms: chronic pain disorders—neurostimulation therapies—human studies—inflammatory phenotyping.

## 3. The Role of Circulating Inflammatory Mediators in Chronic Pain Development

Cytokine and chemokine mediated inflammation is an essential aspect of the pathophysiological processes of pain. Evidence from clinical and pre-clinical data support the notion that pro-inflammatory cytokines including IL-1β, TNF-α, and chemokines such as monocyte chemotactic protein-1 (MCP-1) directly modulate cell activity in different types of neural cells both in the peripheral and central nervous systems (PNS and CNS respectively) [19,20,21,22,23]. In the PNS, topical application of TNF-α to the peripheral axons evoked abnormal spontaneous activity from nociceptive neurons [20]. In a similar manner, TNF-α application to the somata of the DRG neurons *in vitro* provoked abnormal spontaneous activity. Moreover, large myelinated fast-conducting Aβ neurons specifically are excited by topical application to the DRG of TNF-α or by an autologous herniated nucleus pulposus extract [21]. Interestingly, TNF-α can enhance the sensitivity of sensory neurons to the excitation produced by capsaicin, and this enhancement likely is mediated by the neuronal production of prostaglandins [20]. Mechanistically at the level of cellular signals, TNF-α-induced neuronal excitation is mediated by the cAMP-dependent protein kinase (PKA) pathway [20]. The p38 mitogen-activated protein kinase (MAPK) is also involved in TNF-α-induced cutaneous hypersensitivity to mechanical or thermal stimulation [24]. Interestingly, the data from IL-6 knockout mice indicate that IL-6 plays a facilitating role in sympathetic sprouting induced by nerve injury, and its effect on pain behavior is promoted through sympathetic sprouting in the dorsal root ganglia [25]. Consistent with the above data, localized inflammation of the DRG up-regulates a number of pro-inflammatory cytokines including IL1-β, IL-6, TNF-α, HMGB-1, DAMP´s, oxytocin, multiple chemokines, reactive oxygen species, ATP, lipids, and vasoactive amines induces abnormal sympathetic sprouting in the absence of peripheral nerve injury [26]. It suggests a possible correlation between inflammatory responses and sympathetic sprouting, which are two well-known mechanisms implicated in various chronic pain states. This data taken together shows, pro-inflammatory cytokines to be involved in the pathophysiology of the development of inflammatory and neuropathic pain with intense sympathetic nervous system involvement. Hence, specific cytokines and their neutralizing antibodies could be utilized locally or systemically to block pro-inflammatory cytokines for the treatment of chronic pain. These specific antibodies against pro-inflammatory cytokines would act to disrupt the hyperexcitability cycle taking place in the sensory neurons and hence may provide a new, non-NSAR, non-opioid therapeutic approach for the treatment of pathological pain due to inflammation or peripheral nerve injury. Neurostimulation therapies (non-invasive–invasive) targeting different neural targets are successfully being used in clinical pain medicine, but their mode of action and their interaction with cellular and molecular properties of the neuro-immune axis in humans has been poorly investigated, but would probably have a considerable diagnostic and therapeutic yield for pharmacological and non-pharmacological (neurostimulation) approaches (Scheme 1).

### 3.1. Possible Associations between Back Pain (Spino-Nociceptive Traffic) with Immunometabolism

Experimental and in-human studies indicate that obesity and associated diseases such as diabetes or cardiovascular disorders (heart failure, stroke, and cerebral ischemia) can be regarded as a consequence of ongoing chronic inflammation. However, both preclinical and human trials have often been contradictory in assessing the relationships between low back pain and chronic inflammation [27,28,29,30,31,32]. In part, metabolic disorders and related comorbidities have gaped to the development of low back pain indicative of an imbalanced brain–immune communication, which could promote central and peripheral inflammation [32,33,34]. The complex interplay of pro- and anti-inflammatory cytokines (e.g., IL-6, IL-10, IL-13, tumor-necrosis factor (TNF-α), and oxytocin) and other pro- and anti-obesity-relevant peptides, especially leptin (LP), adiponectin (AN), and ghrelin (GH), are thought to evoke the genesis and maintenance of both chronic conditions (pain–metabolic disorder) [28,32,33,34,35,36,37,38,39]. Notably, immunoregulatory IL-10, TNF-α, and IL-1 have been linked to the pathophysiology of multiple chronic disorders including pain, obesity, diabetes, metabolic syndrome, and depression/anxiety, most of which are frequently observed clinically [6,7,34,40].

In this context, a principal member of the adipokine superfamily, leptin (LP), derived from adipocytes, vascular smooth muscle cells, and cardiomyocytes, represents one of the key players in favor of an obese state with studies supporting the relationship between LP expression and body fat mass [34,41,42]. Contrary to IL-10, elevated LP concentration may predict the development of obesity, myocardial ischemia, heart failure, and insulin resistance on the one hand. Clearly, chronic pain is fraught frequently with obesity, but not in general; hence leptin and relatives should be integrated into chronic pain assessment on the other hand as LP drives IL-1 synthesis, which in turn promotes a chronic pain state [34]. Contrary, decreased concentrations of adiponectin (AN), another member of the adipokine family and a neuropeptide solely synthetized by adipocytes, has been suspected of inducing metabolic disorders (e.g., type 2 diabetes, arterial hypertension, and coronary artery disease). Whether concentration changes in circulating AN relate to changes in body weight (body mass index; BMI) or only represent an epiphenomenon is still largely unknown [4,5,6,41,42,43]. AN affects the resistance of insulin receptors (anti-diabetic effect), lipid oxidation, vasodilatation, and anti-arteriosclerosis effects [34,41,42]. To generate an estimated risk profile for metabolic disorders, some studies used the LP–AN ratio (LAR) rather than LP or AN quantification alone. The neuropeptide ghrelin (GH) promotes lipolysis and protects the vascular endothelial architecture (vasoconstriction-suppression) effects. GH is synthesized in the stomach and binds to receptors in the central hypothalamus to stimulate the appetite. Clinical studies have indicated that GH levels are positively correlated with body fat mass. Decreased GH concentrations were detected in obese individuals compared to non-obese controls. Thus, the relationships between chronic low back pain, a very common spinal-nociceptive issue, and immune-driven metabolic disorders appear to be intricately intertwined [44,45,46,47,48,49].

### 3.2. Possible Associations between Migraine (Trigemino-Nociceptive Signaling) with Immunometabolism

There is increasing evidence that migraine, a neurological disorder of the trigeminal-nociceptive system, is associated with the contribution of adipose tissue cellular components of systemic inflammation. Circulating mediators of inflammation involved in the mechanisms of migraine pathology are secreted from adipocytes and adipose tissue-derived immune cells [4]. It is widely recognized, white adipose tissue (WAT), an endocrine-active organ, not only functions as a site of energy storage but also has the capability to exacerbate or reduce systemic inflammation. Within WAT there is a complex interplay between adipocytes involving the balance of synthesis of LP and AN, and the neuro-immune axis to essentially establish physiological homeostasis which, if dysfunctional, leads towards obesity and drives cascades of neuroinflammatory mediators that impact many of the neural circuits relevant to migraine [50,51]. LP (pro-inflammatory), a metabolic marker produced by WAT cells drives COX-2 related pathways of IL-1β in glial cells and neurons of the hypothalamic–pituitary axis. Additional experimental work demonstrated that ghrelin improved light sensitivity, an indicator for migraine-related autonomic features, and evoked behavioral changes in an experimental animal model of head pain [51].

### 3.3. Oxytocin at the Crossroad of Trigeminal-Spinal Pain Transmission and Immunometabolism

Preclinical as well as human studies indicate that the neuropeptide oxytocin impacts pain perception and processing of both the trigemino-nociceptive (e.g., migraine) and the spino-nociceptive (neuropathic pain) systems [52,53,54,55,56,57,58,59,60,61,62,63,64,65,66,67,68]. In addition, oxytocin interacts with metabolic neural circuits and metabolic-related organ systems (liver, muscle, adipose tissue, and pancreas) [53]. The magnocellular neurons located in the hypothalamic nuclei PVN (nucleus paraventricularis) and the SOP (supraoptic nucleus) synthesize oxytocin, which is transmitted in vesicles via anatomical projections to the posterior pituitary lobe, the amygdala, the hippocampus, and the cerebral cortex, ultimately promoting oxytocin release into the systemic blood circulation. A second population of parvocellular oxytocinergic neurons in the PVN projects directly to the brainstem and the spinal cord (dorsal column layers–dorsal root ganglion) without the involvement of the systemic blood circulation [54,55,56]. Based upon available evidence, oxytocin appears to negatively impact nociceptive transmission and signaling in all these areas directly, via on-site neuronal delivery, or indirectly, via the blood circulation, thereby regulating both central and peripheral inflammatory processes.

Experimental animal models and in-human studies have shown positive analgetic effects of oxytocin on cephalgia. After administration of radiographic labeled oxytocin, high concentrations were recorded in the trigeminal nucleus caudalis, the trigeminal ganglia, and the trigeminal branches [57,58]. In addition, several uncontrolled cohort trials observed a meaningful head pain reduction in migraineurs after oxytocin administration. So far, evidence suggests oxytocin may be involved in central sensitization and autonomic dysregulation, conditions frequently observed in migraine phenotypes [59,60,61]. As mentioned, oxytocin projects directly via descending neural routes to specific neuronal cell layers of the spinal cord (dorsal horn laminae) and in addition, is able to bind to receptors of the dorsal root ganglia neurons (DRG) (Scheme 2). This may explain the anti-nociceptive effects of oxytocin in spino-nociceptive circuits elated to pain processing [5,6,7,64,65,66,67,68].

Metabolic disorders frequently co-occur in chronic pain, independently of the pain origin, leading to the question of what role oxytocin may play in metabolic hemostasis under chronic pain conditions. Currently, the hypothalamic–pituitary axis has been recognized as a potential target to restore metabolic function and to impact chronic pain-associated neural transmission, either directly (neuronal) or indirectly (systemic blood circulation). In particular, this becomes an important issue, if clinical phenotypes (pain-metabolism) become difficult to distinguish on a molecular level. It appears that both common and different pathways exist, but this notion is still to be elucidated. Consequently, clinical neurostimulation studies should seek to incorporate metabolic-related markers with the potential to predict chronic pain by diagnostic means. For instance, intranasal administered oxytocin reduced weight, improved insulin sensitivity, and increased pancreatic responsivity in small-scale studies [53]. However, in opposition to what is seen in human trials with apparent conflicting findings, experimental animal studies suggest that oxytocin impacts glucose uptake and lipid utilization in muscle and adipose tissue. This suggests a deterioration in oxytocin signaling may exacerbate obesity and related vascular disorders. 

## 4. Clinical Implications for Neurostimulation Therapies Targeting Chronic Pain Disorders

Possible pathways through which central and peripheral neuromodulation may impact circulating markers of immunometabolism are largely under-investigated in clinical neuromodulation trials. These mechanisms are not likely to be specific for single cytokines, but more likely help control inflammation in quite diverse inflammatory disorders, by activating multiple neuronal networks to control immune cells and thereby minimize the detrimental effects of excessive inflammation seen in chronic pain, dysfunctional metabolism, stroke, epilepsy, and psychiatric disorders. However, the mechanisms of inflammatory action of the applied central and peripheral electrical stimulation modalities (e.g., SCS, DRG, VNS, non-invasive, and invasive brain stimulation) have yet not been fully clarified. A large number of experimental VNS studies have been published which indicate VNS controls the release of multiple cytokines from peripheral macrophages through the neuro-inflammatory reflex involving the nicotinic acetylcholine receptor (nACh) α7. The neuro-inflammatory reflex is in part orchestrated by the dynamic and complex interaction of pro- and anti-inflammatory mediators (e.g., IL-1β, IL-6, IL-8, IL-10, IL-13, interferon-g (IFN-g), TNF-α, and HMGB-1 protein) [69,70,71,72,73]. Combined electrical stimulation of the sciatic nerve and the vagal nerve was found to increase plasma levels of circulating oxytocin as well as touch and pinch indicating the potential of targeted neurostimulation to evoke changes in molecular circuits relevant for pain and metabolism [71]. Given these facts, VNS has been applied for a variety of inflammatory pain disorders (primary headache disorders, rheumatoid arthritis, chronic back pain, neuropathic leg pain, complex regional pain syndrome, chronic pelvic pain, and fibromyalgia) [74].

## 5. In-Human Chronic Pain-Neurostimulation Studies Addressed to Metabolic Molecular Inflammatory Phenotyping (Cervical Non-Invasive VNS, Tonic and BurstDR SCS, DRG-SCS)

To date, there exist a few randomized-controlled and uncontrolled observational neurostimulation human studies, which have quantified peripheral circulating markers of inflammation related to metabolic homeostasis in chronic pain patients [4,5,6,7,75,76,77,78,79] (Table 1).

Most recently, a randomized sham-controlled VNS migraine study [4] found most of the inter-ictal measured cytokines including IL-6, TNF-α, and HMGB1 unchanged before and after VNS as were the metabolic markers leptin, ghrelin, and adiponectin, although serum concentrations were higher in migraine subjects compared to healthy controls. Pro-inflammatory IL-1β plasma concentrations were higher in the sham-VNS group versus the verum nVNS, whereas anti-inflammatory cytokine IL-10 was elevated in migraine patients as compared to healthy subjects but was suppressed three months post-stimulation in both sham and VNS group. These effects were accompanied by a significantly reduced number of severe attacks per month in the nVNS group. Notably, the average BMI was below 25 m^2^/kg, yet nearly 45% of the study cohort suffered from a metabolic-associated disorder (hypertension, cardiac disease) [4]. In a different recent study, significantly increased saliva levels of anti-inflammatory oxytocin and pro-inflammatory IL-1β were detected in the saliva of migraine patients treated before and after VNS treatment, while obesity was present in only 30% of the cohort [52]. In another study including solely healthy subjects, VNS significantly reduced serum IL-1β, TNF- α, IL-6 IL-8, macrophage inflammatory protein [MIP]-1α, and monocyte chemoattractant protein [MCP]-1 demonstrating the anti-inflammatory impact of VNS [80].

Other observational studies and randomized, double-blinded sham-controlled trials have assessed the influence of SCS using tonic waveforms on inflammatory molecular profiles targeting cerebrospinal fluid or skin blister fluid in patients with chronic nociceptive low back and neuropathic leg pain. Inflammatory measures included vascular endothelial growth factor (VEGF), brain-derived neurotrophic factor (BDNF), monocyte chemotactic protein-1 (MCP-1), IL-2, IL-6, IL-12, IL-15, IL-17, TNF-α, interferon-γ (INF- γ), and cellular ratio (Th1-Th2), of which some were changed after conventional, low-frequency SCS (tonic waveform). Although leptin and related metabolic markers were not included, nor were a systematic assessment of weight and/or the existence of metabolic disease performed, some of the measured cytokines and chemokines have been noted in the context of metabolic diseases and depression [75,76,77,78].

In non-obese, failed-back-surgery syndrome patients (FBSS) treated with BurstDR, circulating anti-inflammatory IL-10 was significantly increased after stimulation, while leptin was significantly higher at the baseline with a trend towards lower levels after stimulation. Adiponectin and ghrelin remained unchanged before and after stimulation compared to the healthy control group. Although the study subjects were classified as non-obese according to the BMI, diabetes, hypertension, and cardiovascular disorders were highly prevalent among those subjects with depressive symptoms and disturbed sleep [6,7].

In complex regional pain syndrome (CRPS) and associated comorbidities, a pro-inflammatory state driven by different mediators which results in the complex phenotypes of CRPS may present. In a study of unilateral, targeted dorsal root ganglion stimulation (spine level L-4), improvements were noted in neuropathic pain intensity, sleep quality, and mood in pre-obese CRPS subjects (BMI 29 ± 5.6 kg/m^2^). The choice to study unilateral selective L4 DRG was based on the fact that the knee sensorial innervation is made mainly by this spinal nerve. We believed that if L3 or L5 DRGs were also added, it would have been more difficult to achieve a more precise change, but could not exclude the fact that multisegmental DRG-stimulation would have promoted different changes. Each study subject in this study was noted to have one or more of the following immuno-metabolic clinical disorders: hypertension, diabetes, or cardiac ischemia. On a molecular level, significantly increased serum concentrations of leptin, high-mobility group box 1 (HMGB-1), TNF- α, and IL-6 were quantified before and after L4-DRG stimulation, while IL-1β was significantly increased pre-stimulation, but not after L4-DRG stimulation, and pre-stimulation levels of increased IL-10 were noted to significantly decrease after DRG stimulation. Adiponectin and ghrelin did not differ between CRPS patients and healthy controls at baseline and follow-up [5]. In an extended subgroup assay, the gene expression profile revealed several significantly up- and down-regulated genes. Despite their function for inflammatory host response and neural nociceptive excitability and traffic, some of the genes assessed from the CRPS patients like free fatty acid receptor 2 (FFAR2); interleukin receptor 1 antagonist (ILRN), interleukin 17-F (IL-17F); phospholipase A2, group IIA (PLA2G2); and NADPH oxidase (NOX1) have been associated with metabolic function such as glucose hemostasis, response to starvation, insulin resistance, lipid catabolic function, angiogenesis, blood pressure, and metabolic functions [79]. With regards to CRPS, caution is needed due to the protean nature of CRPS over time and varying phenotypes. CRPS patients remain unique in their presentation and their clinical features. Depending on the time relationship within the inflammatory process, a specific immunological target can change its role by transitioning from the innate to the adaptive immune response. According to the phenotype, the immunological target can vary in role, being either a critical key link in the initiation of the disease or be an inducible marker as the disease progresses associated with a broad variety of physical features such as trophic skin, vascular changes to metabolic factors, bone turn over, trophic skin changes to dystonia [81]. 

This leads to the conclusion that neuromodulation approaches (waveforms) may need to be different (tonic, burst, frequency-regulated) at different times depending upon the effects it can have on chronic pain and relevant comorbidities in balancing the neuro-immune axis (Scheme 1).

## 6. Conclusions and Future Targeted Research

Beyond doubt, this narrative review did not cover the entire spectrum of molecular mediators involved in immunometabolism and inflammation but rather attempted to provide a comprehensive review of inflammatory markers assessed in clinical human neurostimulation trials for chronic pain. Obviously, the pathophysiology of the chronic pain disorders (migraine, CRPS, CLBP, and FBSS) and the approached neural target reported in our review differ fundamentally but may have some overlap in relevant comorbidities such as metabolic disease. In view of the provided data, it appears that molecular profiling of circulating markers of immunometabolism has been under-appreciated and deserves enhanced clinical research attention in the study of patients with chronic pain being treated with neurostimulation therapy. The broad variety of clinical phenotypes observed in chronic pain syndromes and associated comorbidities, of which metabolic-associated disorders are one major representative, points to the urgent need of translational neurostimulation research considering leptin and its related biomarkers of immunometabolism, adiponectin, and ghrelin along with molecular and cellular promoters of neuroinflammation. Hence, a multiplex analysis should be considered via both flow cytometry and mass cytometry techniques to properly determine both extracellular and intracellular changes. These approaches may be complemented by computational tools (computational neurosciences) [82].

To address these translational issues, randomized-controlled neurostimulation trials for chronic pain using a standardized pre-analytic protocol (sample storage, assay, and processing techniques), considering potential bias (age, gender, epigenetic/genetic, and environmental factors) and accompanied by experimental studies are highly warranted.

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
