# Peer review of "Leptin and Associated Mediators of Immunometabolic Signaling: Novel Molecular Outcome Measures for Neurostimulation to Treat Chronic Pain"

_ijms, 2019, doi:10.3390/ijms20194737_

Round 1
Reviewer 1 Report
Manuscript entitled „Leptin and associated mediators of immunometabolic signaling: Novel molecular outcome measures for neurostimulation to treat chronic pain“ provides excellent topic potentially very interesting for readers. However, some revisions are necessary before its publication.
Line 118, authors describe that proinflammatory cytokines are involved in the pathophysiology of pain, however, in the next sentence (line 120) authors mentioned the blocking of antiinflammatory cytokines in the treatment of pain. Please check and correct this discrepancy.
Scheme 1, authors used terms CCL2, MIF, PGE2, NOS, MMPs, M1/M2, neutrophils, Th1/Th2, NK, CD8 cells, B cells, Lipocallin-2, and Progranulin which are not cited in the text. All data mentioned in Figures/Schemes must be supported by references mentioned in the main text or in Figure notes. Moreover, please add the title for this Scheme and also Table 1 in the manuscript.
I strongly suggest to improve article by adding these recent important references into Introduction:
Pain Med. 2019 Jun 1;20(Supplement_1):S58-S68.
Biomed Pharmacother. 2018 Feb;98:424-432.
Transl Neurosci. 2013 Sep;4(3).
Scand J Pain. 2019 Apr 24;19(2):235-244.
Front Psychol. 2018 Nov 29;9:2400.
Medicine (Baltimore). 2017 Mar;96(9):e6130.
After the corrections, the article will meet all requirements for publishing in Int J Mol Sci. I congratulate authors to valuable work!
Author Response
REVIEWER 1
Comments and Suggestions for Authors
Manuscript entitled „Leptin and associated mediators of immunometabolic signaling: Novel molecular outcome measures for neurostimulation to treat chronic pain“ provides excellent topic potentially very interesting for readers. However, some revisions are necessary before its publication.
Line 118, authors describe that proinflammatory cytokines are involved in the pathophysiology of pain, however, in the next sentence (line 120) authors mentioned the blocking of antiinflammatory cytokines in the treatment of pain. Please check and correct this discrepancy.We apologize for having raised confusion and corrected the prase. Corresponding changes were made and marked red lines 120-121.
Scheme 1, authors used terms CCL2, MIF, PGE2, NOS, MMPs, M1/M2, neutrophils, Th1/Th2, NK, CD8 cells, B cells, Lipocallin-2, and Progranulin which are not cited in the text. All data mentioned in Figures/Schemes must be supported by references mentioned in the main text or in Figure notes. Moreover, please add the title for this Scheme and also Table 1 in the manuscript.
Corresponding references were included at the end of the figure title. The title was added for both, table and figure. Changes were made and marked red lines 223-224 and 252-253.
I strongly suggest to improve article by adding these recent important references into Introduction:
Pain Med. 2019 Jun 1;20(Supplement_1):S58-S68.
Biomed Pharmacother. 2018 Feb;98:424-432.
Transl Neurosci. 2013 Sep;4(3).
Scand J Pain. 2019 Apr 24;19(2):235-244.
Front Psychol. 2018 Nov 29;9:2400.
Medicine (Baltimore). 2017 Mar;96(9):e6130.
We thank the reviewer for his valuable hints. The mentioned references were included in the Introduction (Pain Med. 2019 Jun 1;20(Supplement_1):S58-S68; Biomed Pharmacother. 2018 Feb;98:424-432; Scand J Pain. 2019 Apr 24;19(2):235-244; Front Psychol. 2018 Nov 29;9:2400; Medicine (Baltimore). 2017 Mar;96(9):e6130) and in the Discussion (Transl Neurosci. 2013 Sep;4(3). Corresponding changes were made and marked red line 66,71,74-75,78,81,331,371-373,395-406,579-581.
After the corrections, the article will meet all requirements for publishing in Int J Mol Sci. I congratulate authors to valuable work!
Reviewer 2 Report
Abstract: OK . Introduction: line 58-62: Nociceptive pain is a normal response to tissue injury and its associated information is the result of activation and sensitization of pain pathways by multiple inflammatory mediators including leptin, its relatives adiponectin and ghrelin, interleukin (IL) IL1‐β, tumor necrosis factor TNF‐α, IL‐6, high‐mobility group box‐1 HMGB‐1, damage‐associated molecular pattern DAMP ́s, oxytocin, and multiple chemokines, reactive oxygen species, ATP, lipids, and vasoactive amines. . The authors define leptin, its relatives adiponectin and ghrelin, interleukin (IL) IL1‐β, tumor necrosis factor TNF‐α, IL‐6, high‐mobility group box‐1 HMGB‐1, damage‐associated molecular pattern DAMP ́s, oxytocin, and multiple chemokines, reactive oxygen species, ATP, lipids, and vasoactive amines as multiple inflammatory mediators. However, in the Scheme I , the authors contribute these factors to inflammation and chronic pain. Personally, I and many readers may not agree with the authors’ classification, because all these factors, as the authors mentioned in the line 58-62, are multiple inflammatory mediators. THe authors need to define the role(s) of these factors in the field of chronic pain and inflammation carefully and avoid confusing the readers. Besides, I feel the scheme 1 is too simple, and the schem1 needs explanation clearly. . Line 73: CRPS? . Methods: Please define the search years in this study. . 3. In general, the structure of a research paper contains abstract, introduction, methods/materials, results, and discussion. Where is your results? What is your results after you search from internet? How many papers did you enroll in this paper? The inclusion and exclusion criteria and results you should explain clearly in this paper. . 4. It seems that “3. The role of circulating inflammatory mediators in chronic pain development” is like an independent story. However, you put several factors in this section, but the core concept of this section is astray and not systematic. Importantly, if inflammation is your main point, you did not discuss “what cause the inflammation or the increase of circulating inflammatory mediators” in this section. You only descripted the phenomenon of inflammation. The root or basic point of inflammation, which aggravates the circulating inflammation mediators in chronic pain development, is not clearly discussed. . 5. Line 131-133: “Experimental and in‐human studies indicate that obesity and associated diseases such as diabetes or cardiovascular disorders (heart failure, stroke, cerebral ischemia) can be regarded as a consequence of an ongoing chronic, low‐grade inflammation.” This is in part data regarding the obesity, diabetes or cardiovascular disorders. In the method, you mentioned that you were searching “genetic/epigenetic expression”. The diseases such as obesity, diabetes and cardiovascular disorders are no doubt related to several genetic epigenetic expression, but you did not discuss deeply in this part. Would it be reasonable you delete the genetic epigenetic expression in the method, or you would like to discuss more in this section? . 6. Section 3.2 The discussion in the section is too simple. There are several new topics regarding trigemino-nociceptive signaling, such as HCN channel, TRPV channel, and calcium channel, and the novel findings regarding histamine in the pain the author missed in this paper. . 7. 3.3 Oxytocin is the most interesting target in this paper. I would suggest the author add a scheme to show the readers the functions of oxytocin in pain, inflammation and immunomodulation. . 8. The authors have spent a lot of times and space introducing “complex regional pain syndrome” in this paper. If this disease is the main role in this paper. The authors should show your readers how you approach the disease in “inflammation”, “chronic pain”, and “obesity” angle to make a more comprehensive story. . 9. Line 303: why L4, not other levels is so important in the study, the authors did not explain. . 10. Line 231-232 electrical stimulation techniques (SCS, DRG, VNS)? Why not include deep brain stimulation? TMS? Do you know some novel immunomodulation agents, small molecules, and even the IVIG, may have anti-inflammation effects? The authors should add a section regarding the immunomodulation pharmacology to let the readers know the advancing discovery of the pharmatheurtical agents in this field. 11. The conclusion is weak. The authors should highlight the leptin and Novel molecular outcome measures for neurostimulation in the treatment of chronic pain. Too many redundant words about the proinflammatory and inflammatiory detection cannot attract readers to think deeper and innovate readers to extend what they have known and done.Author Response
Reviewer 2
We would like to express our greatest thanks to the reviewer for the effort and the time he put into the review of our manuscript. The authors thank the reviewer and thoroughly considered the comments point by point. Please find a point-to-point revision according to the questions and concerns.
Abstract: OK .
Introduction
line 58-62: Nociceptive pain is a normal response to tissue injury and its associated information is the result of activation and sensitization of pain pathways by multiple inflammatory mediators including leptin, its relatives adiponectin and ghrelin, interleukin (IL) IL1‐β, tumor necrosis factor TNF‐α, IL‐6, high‐mobility group box‐1 HMGB‐1, damage‐associated molecular pattern DAMP ́s, oxytocin, and multiple chemokines, reactive oxygen species, ATP, lipids, and vasoactive amines. . The authors define leptin, its relatives adiponectin and ghrelin, interleukin (IL) IL1‐β, tumor necrosis factor TNF‐α, IL‐6, high‐mobility group box‐1 HMGB‐1, damage‐associated molecular pattern DAMP ́s, oxytocin, and multiple chemokines, reactive oxygen species, ATP, lipids, and vasoactive amines as multiple inflammatory mediators. However, in the Scheme I , the authors contribute these factors to inflammation and chronic pain. Personally, I and many readers may not agree with the authors’ classification, because all these factors, as the authors mentioned in the line 58-62, are multiple inflammatory mediators. THe authors need to define the role(s) of these factors in the field of chronic pain and inflammation carefully and avoid confusing the readers.We regret having raised confusion. Hence an additional section was included in the Introduction describing the interplay between inflammation and chronic pain and pointing to the important fact, that the described molecules have been involved in chronic disorders beyond pain. Corresponding changes were made and marked red.
Besides, I feel the scheme 1 is too simple, and the schem1 needs explanation clearly.
The purpose of this narrative review was to summarizes published human neurostimulation studies for chronic pain, in which molecular profiling of inflammatory markers was performed. A title was added for scheme 1 describing the purpose of the scheme. Indeed, it may appear simplified and the mentioned mediators represent a fraction/puzzle piece of more complex circuits. Hence, future human studies should consider the quantification of multiple cellular and molecular components. If inappropriate, we agree to delete scheme 1.
Line 73: CRPS? .
The abbrevation was spelled out. Corresponding changes were marked red.
Methods:
Please define the search years in this study. .The mssing information was added.
In general, the structure of a research paper contains abstract, introduction, methods/materials, results, and discussion. Where is your results? What is your results after you search from internet?
We agree with the reviewer about the given structure of research papers. With respect to review articles, there exist 3 options (narrative reviews, qualitative systematic reviews, quantitative systematic reviews/meta-analyses). Contrary to a PRISMA based systematic review, there are no acknowledged guidelines for a narrative review. Hence, we structured our narrative review into Introduction – Methods – Main Body (Discussion) – Conclusion. The decision to draft a narrative type of review was mainly routed by the fact that in total 10 papers exist dealing with this important issue and certainly future work is needed in this novel field of neuromodulation for chronic pain. Hence, we did not change the format of our review.
How many papers did you enroll in this paper? The inclusion and exclusion criteria and results you should explain clearly in this paper.
The inclusion criteria was defined in the method section. Corresponding changes were made and marked red. In total, 10 research papers were evaluated and incorporated in this narrative review. To better illustrate relevant studies meeting the inclusion criteria we drafted a table in the first submission and according to reviewer 1, we added a table title, clearly showing the characteristics of the included studies.
It seems that “3. The role of circulating inflammatory mediators in chronic pain development” is like an independent story. However, you put several factors in this section, but the core concept of this section is astray and not systematic. Importantly, if inflammation is your main point, you did not discuss “what cause the inflammation or the increase of circulating inflammatory mediators” in this section. You only descripted the phenomenon of inflammation. The root or basic point of inflammation, which aggravates the circulating inflammation mediators in chronic pain development, is not clearly discussed.
The purpose of this section to generally introduce the role of inflammation in chronic pain, bridging over to the multifunctional aspect of the described inflammatory mediators. The mentioned peptides in this section were chosen according to the published neurostimulation human data for chronic pain. We fully agree with the reviewer, that among the mentioned peptides, there exist a large number of factors influencing the development of chronic pain and co-morbidities. However, future trials in this area should seek to assess a wide range of circulating cytokines/chemokines.
Line 131-133: “Experimental and in‐human studies indicate that obesity and associated diseases such as diabetes or cardiovascular disorders (heart failure, stroke, cerebral ischemia) can be regarded as a consequence of an ongoing chronic, low‐grade inflammation.” This is in part data regarding the obesity, diabetes or cardiovascular disorders. In the method, you mentioned that you were searching “genetic/epigenetic expression”. The diseases such as obesity, diabetes and cardiovascular disorders are no doubt related to several genetic epigenetic expression, but you did not discuss deeply in this part. Would it be reasonable you delete the genetic epigenetic expression in the method, or you would like to discuss more in this section? .
The search term genetic/epigenetic was deleted from the method section.
Section 3.2 The discussion in the section is too simple. There are several new topics regarding trigemino-nociceptive signaling, such as HCN channel, TRPV channel, and calcium channel, and the novel findings regarding histamine in the pain the author missed in this paper.
In summary, these are great ideas from the reviewer. The markers identified in the current studies cannot accomplish this, but it is feasible to design novel longitudinal studies in which circulating biomarkers such as HCN channel, TRPV channel, calcium channel and histamine pathways could be used to assess the potential of immunometabolic markers in the treatment of chronic pain disorders. However, the authors felt, that this probably outside the scope of this narrative review, but will certainly consider to quantify the mentioned pathways in future trials.
3 Oxytocin is the most interesting target in this paper. I would suggest the author add a scheme to show the readers the functions of oxytocin in pain, inflammation and immunomodulation. .
The authors have spent a lot of times and space introducing “complex regional pain syndrome” in this paper. If this disease is the main role in this paper. The authors should show your readers how you approach the disease in “inflammation”, “chronic pain”, and “obesity” angle to make a more comprehensive story.
CRPS was one pain disorders among others mentioned in our review, hence we shortened this section to the relevant aspects. Corresponding changes were marked red.
Line 303: why L4, not other levels is so important in the study, the authors did not explain. .
The choice to study unilateral selective L4 DRG was based on the fact that the knee sensorial innervation is made mainly by this spinal nerve. We believed that if L3 or L 5 DRGs were also added, it would have been more difficult to achieve a more precise change. The authors fully agree and acknowledge this important point as the sensory input is processed multisegmental rather than unisegmental. We added this explanation and changes were marked red
Line 231-232 electrical stimulation techniques (SCS, DRG, VNS)? Why not include deep brain stimulation? TMS?
The authors fully agree with reviewer and additional neurostimulation approaches considering both have been included.
Do you know some novel immunomodulation agents, small molecules, and even the IVIG, may have anti-inflammation effects? The authors should add a section regarding the immunomodulation pharmacology to let the readers know the advancing discovery of the pharmatheurtical agents in this field.
Pharmacological agents encompass an impact on the neuro-mmune axis, no doubt. However, the review targeted electrical neuromodulation, hence we think, that an additional section addressed to this issue would be outside the scope and probably would deserve a seperate review.
The conclusion is weak. The authors should highlight the leptin and Novel molecular outcome measures for neurostimulation in the treatment of chronic pain. Too many redundant words about the proinflammatory and inflammatiory detection cannot attract readers to think deeper and innovate readers to extend what they have known and done.
We thank the reviewer to share his concerns with us. However, in this issue, we think, that the conclusion points to the right future direction in this novel and expanding field of neurostimulation. Furthermore, according the to the available literature immunometabolism profling is in the beginning in neurostimulation and deserves enhanced attention.
Reviewer 3 Report
Kinfe et al. provide the current knowledge of inflammatory markers assessed in clinical human neurostimulation trials for chronic pain. The structure of the manuscript is well composed. Well-design table and figure enrich the text. Overall, the entire article is of interest. However, there are several points that remain to be addressed, in my opinion.
The authors should better explain what is characterized by the low-grade inflammation state. The authors state that adipocytokines are pro‐ and anti‐obesity‐relevant peptides, however, mounting evidence indicate that they are also crucial humoral factors and they are considered to have mainly pro-, anti-inflammatory or immunoregulatory properties. This should be clarified in the context of the article subject. MCP-1 is also referred to as the chemokine CCL2; the same in case of MIP-1 or IL-8. Both terms should be used to dispel doubts for readers. There are some grammar or word errors, for instance, line 168 "There is increasing evidence migraine..."; line 183 "Preclinical as well as human studies support the analgetic properties of...". The manuscript would benefit from review by a native English speaker. In the entire text, there are incomprehensible cross-words, e.g., lines 52 and 82. What does the last column of Table 1 mean? This is not clear to the reader.
Author Response
REVIEWER 3
Kinfe et al. provide the current knowledge of inflammatory markers assessed in clinical human neurostimulation trials for chronic pain. The structure of the manuscript is well composed. Well-design table and figure enrich the text. Overall, the entire article is of interest. However, there are several points that remain to be addressed, in my opinion.
We thank the reviewer for his time, the consideration of our manuscript and his valuable hints, which we addressed point-by-point.
The authors should better explain what is characterized by the low-grade inflammation state.The term chonic low-grade inflammation appears to be misleading. Chronic inflammation is better suited to characterize pathophysiological features of chronic pain on a molecular level. Hence, the term chronic low-grade inflammation was replaced by chronic inflammation. Corresponding changes were made and marked red see line 146-148.
The authors state that adipocytokines are pro‐ and anti‐obesity‐relevant peptides, however, mounting evidence indicate that they are also crucial humoral factors and they are considered to have mainly pro-, anti-inflammatory or immunoregulatory properties. This should be clarified in the context of the article subject.
We fully agree and additional sections were added and marked red see line 63-68, line75-79, 154-156,
MCP-1 is also referred to as the chemokine CCL2; the same in case of MIP-1 or IL-8. Both terms should be used to dispel doubts for readers.
We deeply apologize as we did get the section, which deserves revision. We would be thankful for additional hints to appropriately address the reviewer´s concerns.
There are some grammar or word errors, for instance, line 168 "There is increasing evidence migraine..."; line 183 "Preclinical as well as human studies support the analgetic properties of...". The manuscript would benefit from review by a native English speaker. In the entire text, there are incomprehensible cross-words, e.g., lines 52 and 82.
We regret to have overlook grammatical errors although the manuscript has been edited prior to submission by a native english speaker. However, we underwent a careful revision oft he entire manuscript in order to provide a correct syntax and grammar.
What does the last column of Table 1 mean? This is not clear to the reader.
The last column intended to present the observation period oft he published studies. Follow-up was replaced by treatment duration to underpin the generally short-term follow----up. This fact is of improtance, as metabolic-associated changes may appear after a longer time period as presented by the published studies.
Round 2
Reviewer 1 Report
Authors implemented all my suggestions, the paper was significantly improved.
Reviewer 2 Report
accepted